# LASP1, CERS6, and Actin Form a Ternary Complex That Promotes Cancer Cell Migration

**DOI:** 10.3390/cancers15102781

**Published:** 2023-05-16

**Authors:** Atsuko Niimi, Siripan Limsirichaikul, Keiko Kano, Yasuyoshi Mizutani, Toshiyuki Takeuchi, Patinya Sawangsri, Dat Quoc Tran, Yoshiyuki Kawamoto, Motoshi Suzuki

**Affiliations:** 1Department of Molecular Oncology, Fujita Health University, Toyoake 470-1192, Japan; 2Department of Biopharmacy, Faculty of Pharmacy, Silpakorn University, Nakhon Pathom 73000, Thailand; 3Institute of Transformative Bio-Molecules (WPI-ItbM), Nagoya University, Chikusa, Nagoya 464-8602, Japan; 4Department of Biomedical Sciences, College of Life and Health Sciences, Chubu University, Kasugai 487-8501, Japan

**Keywords:** CERS6, LASP1, metastasis, lung cancer, cell migration, lamellipodia, actin, ceramide, LIM domain, cytoskeleton

## Abstract

**Simple Summary:**

CERS6 is known to be associated with metastasis and poor prognosis in non-small cell lung cancer (NSCLC) patients because of C16 ceramide production, though the underlying mechanism remains largely unelucidated. In this study, CERS6 binding partners were identified by co-immunoprecipitation, as well as liquid chromatography and tandem mass spectrometry analysis. LASP1, one of the leading candidates, was found to play a role in cell migration, while CERS6 and LASP1 were shown to co-localize on lamellipodia and interact with actin. Silencing of CERS6 and/or LASP1 led to reduced levels of lamellipodia formation and cell migration. Furthermore, interaction of CERS6 or LASP1 with actin was dramatically decreased when the partner was depleted. Based on these findings, it is proposed that LASP1–CERS6 interaction promotes cancer cell migration.

**Abstract:**

CERS6 is associated with metastasis and poor prognosis in non-small cell lung cancer (NSCLC) patients through d18:1/C16:0 ceramide (C16 ceramide)-mediated cell migration, though the detailed mechanism has not been elucidated. In the present study, examinations including co-immunoprecipitation, liquid chromatography, and tandem mass spectrometry analysis were performed to identify a novel binding partner of CERS6. Among the examined candidates, LASP1 was a top-ranked binding partner, with the LIM domain possibly required for direct interaction. In accord with those findings, CERS6 and LASP1 were found to co-localize on lamellipodia in several lung cancer cell lines. Furthermore, silencing of CERS6 and/or LASP1 significantly suppressed cell migration and lamellipodia formation, whereas ectopic addition of C16 ceramide partially rescued those phenotypes. Both LASP1 and CERS6 showed co-immunoprecipitation with actin, with those interactions markedly reduced when the LASP1–CERS6 complex was abolished. Based on these findings, it is proposed that LASP1–CERS6 interaction promotes cancer cell migration.

## 1. Introduction

Metastasis is a hallmark of cancer, as well as a leading cause of aggressive phenotype development and poor prognosis of affected patients. Elucidation of the cancer metastasis process is an urgent need, along with the development of preventive and therapeutic strategies. The present study focused on lung cancer, a leading cause of death worldwide.

It has been reported that ceramide synthases (CERSs) and levels of their enzymatic products are associated with anti-cancer drug sensitivity [1,2,3,4,5]. Despite their apoptotic functions, it has also been found that various cancers are associated with a high level of ceramides or CERS expression in cancer tissues [6,7]. In accord with those reports, we previously showed that CERS6 overexpression in non-small cell lung cancer (NSCLC) was associated with poor prognosis and distant lymph node metastasis [8]. Additional analyses revealed that CERS6 stimulated lamellipodia formation and cell migration via C16 ceramide-dependent RAC1-PKCξ activation, while *miR-101* [8] and the transcription factors CEBPγ and YBX1 [9] regulated expression levels.

Other groups have also reported the cancer-promotion roles of CERS6. For example, CERS6 was found to suppress endoplasmic reticulum (ER)-stress mediated apoptosis in human head and neck squamous cell carcinomas (HNSCCs) [10]. In an analysis of breast cancer, CERS6 overexpression conferred chemoresistance, while its inhibition significantly reduced growth, migration, and survival of cancer cells [11]. Another study also showed that ceramides are required for the migration activity of A549 cells by negative regulation of ceramide kinase [12].

Structural analysis of LASP1, an actin-binding protein, has revealed that it is composed of an N-terminal LIM domain with two zinc finger motifs, followed by two central actin-binding nebulin repeats termed R1 and R2 [13]. These domains are flanked by a linker region and C-terminal SH3 domain, which are known to bind to several proline-rich segment proteins. It has also been shown that the actin-binding domains of LASP1 mediate a direct interaction between LASP1 and actin at cell membrane extensions [14]. The binding of LASP1 to actin stress fibers is mediated via its interaction with the protein palladin, which binds to the SH3 domain of LASP1 [15]. Other binding partners of the LASP1 LIM domain are chemokine receptor 2 (CXCR2) and cadherin-11/catenin, which have important roles for the migration of epithelial and immune cells, as well as during the processes of angiogenesis, inflammation, wound healing, and atherosclerosis [16,17].

In the present study, co-immunoprecipitation (IP), liquid chromatography, and tandem mass spectrometry (LC-MS/MS) examinations were conducted, which showed interaction of CERS6 with LIM and SH3 domain protein 1 (LASP1), an actin-binding protein. Subsequent analyses also revealed a critical role of this complex for the promotion of cancer cell migration.

## 2. Materials and Methods

### 2.1. Cell Lines and Cultures

A549, NCI-H460-LNM35 (LNM35) [18], and BEAS-2B were provided by Dr. Takashi Takahashi at the Nagoya University School of Medicine. A549 and LNM35 cells were cultured and maintained in RPMI1640 supplemented with 5% FBS (Gibco). The immortalized lung epithelial cell line BEAS-2B was used for CERS6 overexpression, as previously described [19]. For some experiments, cells were cultured under an “induction condition”, in which they were cultured in RPMI1640 with N2 supplemented medium (Gibco) for 48 h, followed by RPMI1640 supplemented with 10% FBS for 20 h. All cell lines were determined to be free from mycoplasma infection.

### 2.2. Antibodies and Chemical Reagents

Anti-CERS6 antibody (H00253782-M01, clone 5H7) was purchased from Abnova. Anti-LASP1 (ab117806) and anti-histone H3 (ab18521) antibodies were from Abcam. Anti-RAC1 antibody (05-389, clone 23A8) was from Merck-Millipore. Anti-β actin (A5441), anti-CERS2 (HPA027262), and anti-FLAG (F7425) antibodies were from Sigma-Aldrich (St. Louis, MO, USA). Anti-HA tag antibody (clone TANA2, M180-3) was from MBL Life Science. Normal mouse IgG, normal rabbit IgG, anti-mouse IgG, and anti-rabbit IgG conjugated with HRP were from Cell Signaling Technology. Anti-mouse IgG and anti-rabbit IgG conjugated with Alexa 488 or 568 were from Invitrogen. cOmplete™ EDTA-free Protease Inhibitor Cocktail (11873580001 Roche) was from Merck. C16 ceramide was from Avanti Polar Lipids. Phalloidin-Rhodamine X conjugated (phalloidin, 165-21641) was from FujiFilm Wako Pure Chemical.

### 2.3. siRNA Transfection

For RNAi, 20 nM of each siRNA duplex (Sigma-Aldrich) (Appendix A) targeting CERS6 or LASP1, or the same concentration of the negative control (Mission siRNA universal negative control #2, Sigma-Aldrich) (siCTRL), was transfected using Neon™, according to the manufacturer’s instructions, and cultured for 72 h.

### 2.4. Plasmid Construction and Transfection

CERS6 pcDNA3 was constructed as previously described [20]. HA-tag was PCR-amplified (for PCR primers, see Appendix A) and inserted at the CERS6 C-terminus (CERS6-HA-pcDNA3). LASP1-FLAG-pCMV was purchased (HG14046-CF, Sino Biological Japan, Inc). A truncated mutant series of LASP1 was constructed using PCR primer sets (Appendix A). For overexpression of full-length or truncated mutants, cells were transfected with each construct using Neon™, then cultured for 24 h.

### 2.5. Co-immunoprecipitation, Liquid Chromatography, and Tandem Mass Spectrometry (IP and LC-MS/MS)

IP was performed using a Dynabeads™ Co-Immunoprecipitation Kit (Invitrogen 14321D), according to the manufacturer’s protocol. Protein complexes were eluted by boiling in buffer containing 2% SDS, 10% glycerol, 62.5 mM Tris-HCl (pH 6.8), 0.01% bromophenol blue, and 5% 2-mercaptoethanol. Supernatants were collected and subjected to sodium dodecyl sulphate-polyacrylamide gel electrophoresis (SDS-PAGE). After in-gel digestion was performed [21], peptides were analyzed by liquid chromatography–tandem mass spectrometry (for details, see Appendix A).

Based on the values for abundance ratio (anti-CERS6 antibody-bound samples vs. those from anti-IgG antibody), co-precipitated protein candidates were sorted (Appendix A). The experiments were performed twice.

### 2.6. In Vitro Motility and Complementation Assays

Motility assays were performed as previously reported [8]. Briefly, siRNA-transfected LNM35 or A549 cells were cultured under the induction condition. Cells were collected using 5 mM EDTA in PBS and resuspended in RPMI1640 supplemented with 0.1% FBS, then 500 μL was used for seeding at a density of 2 × 10^5^ cells/mL into the upper chambers and incubation was performed for 16–24 h. Then, the bottom side of the upper chamber was fixed with 70% ethanol and staining was performed with Giemsa’s azur-eosin-methylene blue (Merck-Millipore, 109204). For quantification, cell numbers were counted under 200× magnification. These experiments were performed at least three times, and average and SD values were determined. For complementation analysis, 1 μM of C16 ceramide was added to both the induction condition and the RPMI1640 supplemented with 0.1% FBS media.

### 2.7. Quantitative RT-PCR

Total RNA was extracted using an miRNeasy kit (Qiagen, 217084), followed by cDNA synthesis using a SuperScript™ VILO™ cDNA Synthesis Kit (Thermo Fisher Scientific, Waltham, MA, USA). The reaction mixture (10 μL) containing 2 μL of VILO Reaction Mix, 1 μL of SuperScript Enzyme Mix, and 500 ng of cellular RNA was incubated at 25 °C for 10 min, 42 °C for 60 min, and 85 °C for 5 min. SYBR green quantitative RT-PCR analysis was performed using a QuantiTect SYBR Green PCR kit (Qiagen, 204143) with a Rotor Gene 3000 system (Corbett Research), with some modifications, as previously reported [22]. Briefly, a 20 μL reaction mixture containing an equal amount of cDNA, 0.3 μM each of forward and reverse primers (Appendix A), and 10 μL of PCR Master Mix was used. CERS6, LASP1, and internal control gene 18S ribosomal RNA were amplified for 45 cycles at 94 °C for 10 s, 55 °C for 30 s, and 72 °C for 30 s. Ct values were normalized to those of 18S (ΔCt), then average ΔΔCt values were calculated by normalization to the ΔCt value of siCTRL-treated cells, as previously described [23]. Experiments were performed three times and average values with SD were determined.

### 2.8. Ceramide Quantitation

siRNA-treated LNM35 and A549 cells were cultured under the induction condition for 72 h. The lipid fraction of each cell line was then extracted using the Bligh–Dyer extraction method with d18:1/C17:0-ceramide as the internal standard [8,9]. Ceramide analysis was performed using an ACQUITY Ultra Performance LC system (Waters) with a 4000 QTRAP LC/MS/MS device (AB Sciex). High-performance liquid chromatography was performed using gradient mode with water/0.2% formic acid and 60% acetonitrile/40% isopropanol/0.2% formic acid solutions with a conventional ODS column (Cadenza CW-C18, 150 × 2 mm). Mass spectrometry was performed in positive ion mode with an electrospray ionization source [8].

### 2.9. Co-Immunoprecipitation and Western Blotting (IP-WB)

IP-WB examinations were performed using a Dynabeads co-IP kit, as described above in Section 2.5, with some modifications. Briefly, cells were harvested, lysed, and sonicated in IP-lysis buffer (25 mM Tris HCl pH 7.4, 150 mM NaCl, 1 mM EDTA, 1% NP-40, 1% glycerol, protease inhibitor cocktail). The antibody against the target protein was coupled with magnetic beads overnight before incubation in the cell lysate with the beads for 2 h at 4 °C. After washing the beads with IP-lysis buffer 3 times, the co-precipitated protein complex was eluted and subjected to SDS-PAGE. The following antibodies against proteins were used for WB: anti-CERS6 (dilution 1:1000), anti-LASP1 (dilution 1:2000), anti-HA (dilution 1:1000), anti-FLAG (dilution 1:4000), anti-β actin (dilution 1:10,000), and anti-histone H3 (dilution 1:50,000).

For experiments performed to detect LASP1– or CERS6–actin interaction, cells were cultured under the induction condition. IP buffer containing 20 mM Tris HCl (pH 8.0), 150 mM NaCl, 1% triton X-100, 10% glycerol, and a protease inhibitor cocktail was used for both cell lysis and beads washing.

In some experiments, crosslinking immunoprecipitation (CLIP) was performed to detect interaction between CERS6 and LASP1. A Dynabeads co-IP kit was used as described above, with modification of the cell preparation step. Briefly, CERS6-HA-overexpressed cells were incubated with 1% formaldehyde in RPMI1640 for 5 min at room temperature (RT); then, the reaction was quenched by 125 mM glycine solution for 5 min, followed by washing with PBS twice. Under both conditions, co-precipitated protein complexes were eluted, and subjected to SDS-PAGE and WB analyses.

### 2.10. Immunofluorescent Staining

To visualize the localization of LASP1, and/or CERS6, and lamellipodia 2 × 10^5^ cells were seeded into a 3.5 cm dish with an 18 × 18 mm coverslip (Matsunami, C218181) cultured under the induction condition, and then fixed in PBS buffer containing 3% paraformaldehyde and 2% sucrose at RT for 10 min. After washing with PBS, the cells were permeabilized with 0.2% Triton X-100 in PBS for 2 min, followed by washing with PBS. Cells were then treated with the primary antibody against LASP1 (dilution 1:2000) or CERS6 (dilution 1:200) at RT for 2 h, washed with PBS, and treated with the secondary antibody conjugated with Alexa Flour 488 or Alexa Flour 568 (dilution 1:400) at RT for 1 h. To analyze lamellipodia formation, cells were treated with phalloidin (dilution 1:20,000). While a previous publication defined lamellipodia as an RAC1-positive lamellipodia [8], nearly identical staining patterns between those obtained with phalloidin and those obtained with RAC1 were verified with greater than 96% consistency (Appendix A). Coverslips were mounted with Fluoromount (Diagnostic Biosystems), and visualized under a fluorescent microscope (Olympus BX51, DP71) at magnifications of 400× and 1000×. For BEAS-2B, a confocal laser microscope (Zeiss LSM-710, Zen software) was used for analysis. To assess the degree of co-localization, Pearson’s coefficient for the region of interest was calculated using the Imaris software package, version 9.8.2 (Oxford Instruments, Abingdon, UK).

### 2.11. Statistics

For the results of each experiment, values are expressed as average ± SD, while n is used to represent the number of independent experiments. For paired samples, data were analyzed for statistical significance using Welch’s *t*-test and one-way ANOVA, with the Holm–Sidak test for paired and multiple comparisons, respectively. pvalues less than 5% were regarded as significant.

## 3. Results

### 3.1. LASP1 Is a CERS6 Binding Partner

Metastasis is associated with aggressive cancer phenotypes and leads to poor prognosis. To identify proteins that promote this potentially mortal condition, immunoprecipitation using an anti-CERS6 antibody was performed, followed by LC-MS/MS analysis. In addition to the high level of CERS6 and the dimerization partner CERS5, LASP1 showed the highest abundance ratio in each of two independent analyses (Figure 1A, Appendix A), suggesting it as the leading candidate for CERS6 binding. Of those, the present study focused on LASP1, because that protein has been shown to have actin-binding activity and overexpression in various types of cancer specimens [24,25,26]. Additionally, a physical interaction between CERS6 and LASP1 was detected in IP-WB analyses. LASP1 was found to be co-precipitated with CERS6 by use of either the anti-CERS6 or anti-HA antibody in CERS6-HA-overexpressed cells (Figure 1B). Conversely, IP-WB findings showed that CERS6 was co-precipitated with LASP1 (Figure 1C).

To determine whether such an interaction also occurs in living cells, immunocytochemical analysis was performed. In the CERS6-high expression cell lines LNM35 and A549, both CERS6 and LASP1 were, at least in part, co-localized on lamellipodia, while their staining patterns showed far less overlapping in other subcellular regions (Figure 1D). Similar results were noted when the CERS6-low expression cell line BEAS-2B was used for analysis with overexpression of CERS6.

### 3.2. LIM Domain of LASP1 Required for CERS6 Interaction

Truncations were introduced into the LASP1-FLAG construct to determine the CERS6 binding domain. In the A549 cell line, truncated LASP1-FLAG mutants 1 to 4 were overexpressed and immunoprecipitated using an anti-FLAG antibody. As a result, all four were successfully co-precipitated with endogenous CERS6 (Figure 2A,B).

A fifth mutant lacking an N-terminal LIM domain was also constructed (Appendix A), though it was not captured by the anti-FLAG antibody (Appendix A). For this reason, instead of using the fifth mutant, an interaction was confirmed by performing another immunoprecipitation assay from the opposite direction. CERS6-HA and each one of the truncated LASP1-FLAG mutants were co-expressed, and immunoprecipitated with an anti-HA antibody. All of the LASP1 mutants carrying LIM were found to be co-precipitated with CERS6-HA (Figure 2C). These results indicated that the LIM domain is required for CERS6 interaction.

### 3.3. LASP1 and CERS6 Coordinate to Promote Cell Migration

LASP1 was initially discovered in a search of a breast cancer metastasis library and then reported to be involved in cell invasion, migration, and metastasis [24,25,27]. CERS6 is employed for a similar pathological phenotype [8]; thus, it is feasible that in addition to the physical interaction, these genes functionally coordinate and promote cancer metastasis. To investigate this possibility, cell migration and lamellipodia formation efficiency under a LASP1 knock-down condition were examined. The results showed that LASP1 silencing by use of two independent sequences significantly suppressed LNM35 cell migration activity (Figure 3A,B), while a major part of that suppression was rescued by the addition of C16 ceramide. No statistically significant differences between the effects of siCTRL with or without C16 ceramide were noted. Furthermore, LASP1 suppression was found to decrease lamellipodia formation in LNM35 cells, though the level of reduction was not as high as that induced by CERS6 silencing (Figure 3C,D).

Similar results were obtained with A549 cells that showed migration activity reduced by LASP1 silencing (Appendix A). Additional findings indicated that with a double knock-down of CERS6 and LASP1, the activity was similar to that seen with CERS6 silencing. Furthermore, suppression of CERS6 or LASP1 led to a decreased number of lamellipodia (Appendix A). Additionally, in contrast to the significant effects on cell migration activity, silencing seemed to have a scant effect on cell viability (Appendix A).

### 3.4. Complex Formation May Alter LASP1 and CERS6 Levels in LNM35 Cells

Additional examinations showed that LASP1 and CERS6 modulated protein expression levels of the binding partners in LNM35 cells. LASP1 silencing upregulated CERS6 expression, while CERS6 silencing upregulated the expression of LASP1 (Figure 4A and Appendix A). Increased protein amounts in LNM35 may not be the consequence of transcriptional regulation, as mRNA remained at similar levels (Appendix A).

When upregulated, CERS6 protein seemed to remain enzymatically intact, because under the LASP1 silencing condition the C16 ceramide level was also increased, while C24 and C24:1 ceramides, CERS2 products, remained at similar levels (Figure 4B and Appendix A). These results suggest the presence of a feedback pathway in LNM35, though this phenotype was not clearly shown in A549 (Appendix A, see Discussion).

### 3.5. Complex Formation Facilitates LASP1– and CERS6–Actin Interaction

Previous studies have found that LASP1 is an F-actin-binding partner [14,28,29]. In accord with those reports, the present study revealed LASP1– and CERS6–actin stress fiber co-localizations in LNM35 and A549 (Figure 5A and Appendix A). To elucidate the mechanisms that link CERS6 functions with lamellipodia formation, a biochemical assay was conducted to evaluate the interaction between LASP1 and soluble actin molecules. Interestingly, under the CERS6 silencing condition, the amount of LASP1-bound β actin was markedly decreased (Figure 5B), while it was partially recovered by the addition of C16 ceramide (Figure 5C). Furthermore, CERS6 was also co-precipitated with actin in a LASP1-dependent manner (Figure 5D). Together, these results suggest that CERS6 and/or C16 ceramide are required for the formation of a complex between LASP1 and β actin, and that CERS6 may form a ternary complex resulting in efficient lamellipodia formation and cancer cell migration.

## 4. Discussion

LASP1, LIM, and SH3 domain protein 1, a cytoskeleton-associated protein involved in cancer cell migration and invasion, were initially identified in lymph nodes of human patients with breast cancer metastasis [27]. Subsequently, this protein was found to be highly expressed in a variety of tissues with a malignant tumor, such as cases of lung, pancreas, colorectal, hepatocellular, and bladder [24,30,31,32] cancer. Furthermore, a high level of expression has been shown to be associated with aggressive phenotypes and poor prognosis in renal [33] and breast [34] cancer patients.

CERS6 (ceramide synthase 6) is an enzyme that regulates ceramide synthesis, a component of the cell membrane that has roles in cell signaling and apoptosis. Actin is a protein critical for formation of the cytoskeleton, a network of fibers that provides structural support to cells. The present results provide evidence that CERS6, LASP1, and actin form a ternary complex in various lung cancer cell lines. Since both proteins were found on actin stress fiber (Figure 5A and Appendix A), and considering the physical complex made with soluble β actin (Figure 5B), it is speculated that they have an ability to interact with both G- and F- actin, and facilitate cancer cell migration (Figure 6). This is thought to occur by regulation of the dynamics of the cytoskeleton, which allows cancer cells to move and invade surrounding tissue.

In this regard, ternary complex formation may be required to supply a certain amount of C16 ceramide to promote the processes of actin stress fiber assembly and/or disassembly. This proposed model is consistent with clinical evidence showing that both CERS6 and LASP1 are upregulated in different types of cancer, and that high expression is associated with poor prognosis [8,25,26,33].

Regarding crosstalk between CERS6 and LASP1 protein levels, an apparent upregulation was observed in LNM35 but not A549 cells (Figure 4, Appendix A). Since co-localization was detected only in the subcellular structure lamellipodium (Figure 1D), and lamellipodia formation efficiency in A549 was ~3-fold lower than that in LNM35 (Figure 3C and Appendix A), it is possible that the present WB analysis using whole cell extracts was not adequately sensitive to detect the phenotype in A549.

When considering anatomical factors, the LIM domain of LASP1 might be required for interaction with CERS6 protein. In previous studies, LASP1 has been shown to interact with a series of other structural proteins, including F-actin [14,35], Zyxin [36], Palladin [15], Krp1 [37], and vimentin [38], through the R1/R2 and SH3 domains. Therefore, it is feasible that these cytoskeleton-related complexes further assist the CERS6-dependent migration process. While the present results obtained with a series of LASP1 mutants suggest an interaction between the LIM domain and CERS6, we intend to perform another study of LASP1 with a mutated LIM domain in the future before making a definitive conclusion.

Interestingly, disruption of the complex by either LASP1 or CERS6 silencing resulted in upregulation of each binding partner level (Figure 4A). Although precise mechanisms are yet to be elucidated, these results suggest the presence of CERS6- and LASP1-specific feedback mechanisms that maintain active actin dynamics and subsequent lamellipodia formation.

Taken together, the findings presented here suggest a novel model of metastasis promotion. Nevertheless, additional analyses are needed to fully understand the mechanism by which this ternary complex promotes cancer cell migration and its potential as a target for anti-cancer therapy.

## 5. Conclusions

The present IP and LC-MS/MS analysis findings identified LASP1 as a CERS6 binding partner, and also identified the requirement of the LIM domain of LASP1 for the physical interaction. In addition to their interaction, LASP1 and CERS6 were found to coordinate in the promotion of cell migration and lamellipodia formation. Thus, it is proposed that these proteins function together to contribute to the cell migration process. In addition, LASP1 and CERS6 were revealed to interact with β actin, with the interaction of each decreasing with the depletion of the partner, while C16 ceramide partially rescued the LASP1–actin interaction under a CERS6 silencing condition. It is considered that the ternary complex of LASP1, CERS6, and actin may contribute to lamellipodia formation and cancer cell migration.

## Figures and Tables

**Figure 1 cancers-15-02781-f001:**
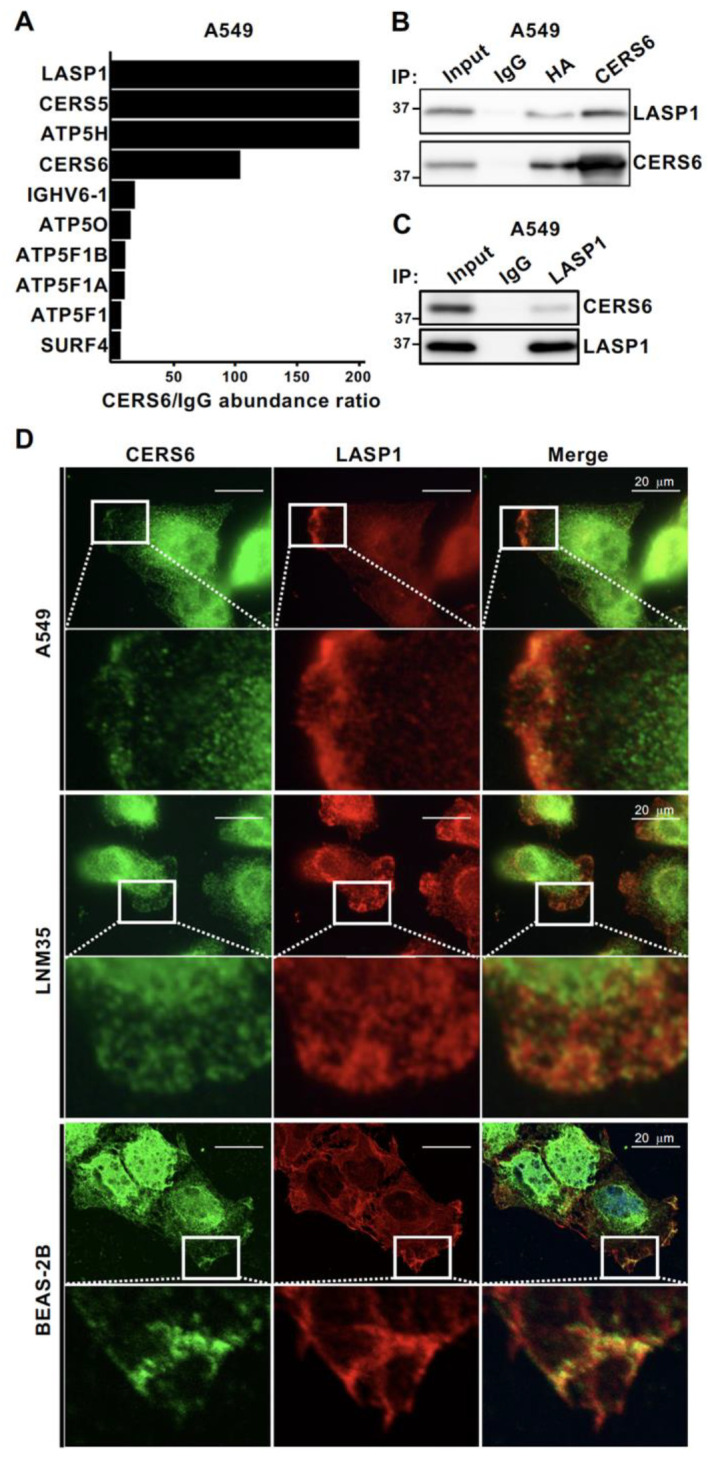
LASP1 is a CERS6 binding partner. (**A**) CERS6-binding proteins identified by IP with the anti-CERS6 antibody and LC-MS/MS analysis. The top ten candidates based on abundance ratio (anti-CERS6 antibody-bound samples vs. those from anti-IgG antibody) are listed. (**B**) CLIP-WB with anti-CERS6 or anti-HA to show LASP1 interaction with CERS6. A549 cells with overexpression of CERS6-HA were incubated with 1% formaldehyde in RPMI1640, then the reaction was quenched using 125 mM of a glycine solution. Cells were harvested, lysed, and sonicated, then IP was performed using a Dynabeads co-IP kit, after which co-precipitated protein complexes were eluted, and subjected to SDS-PAGE and WB analyses. (**C**) IP-WB with anti-LASP1 to show CERS6 interaction with LASP1. After A549 cells were harvested and lysed, IP-WB was performed using a Dynabeads co-IP kit (see Materials and Methods). (**D**) Immunofluorescent staining of LASP1 and CERS6 at lamellipodia of A549, LNM35, and CERS6-overexpressed BEAS-2B cells. The Pearson’s coefficient values for A549, LNM35, and CERS6-overexpressed BEAS-2B were 0.702, 0.789, and 0.812, respectively. Original western blots are presented in Appendix A.

**Figure 2 cancers-15-02781-f002:**
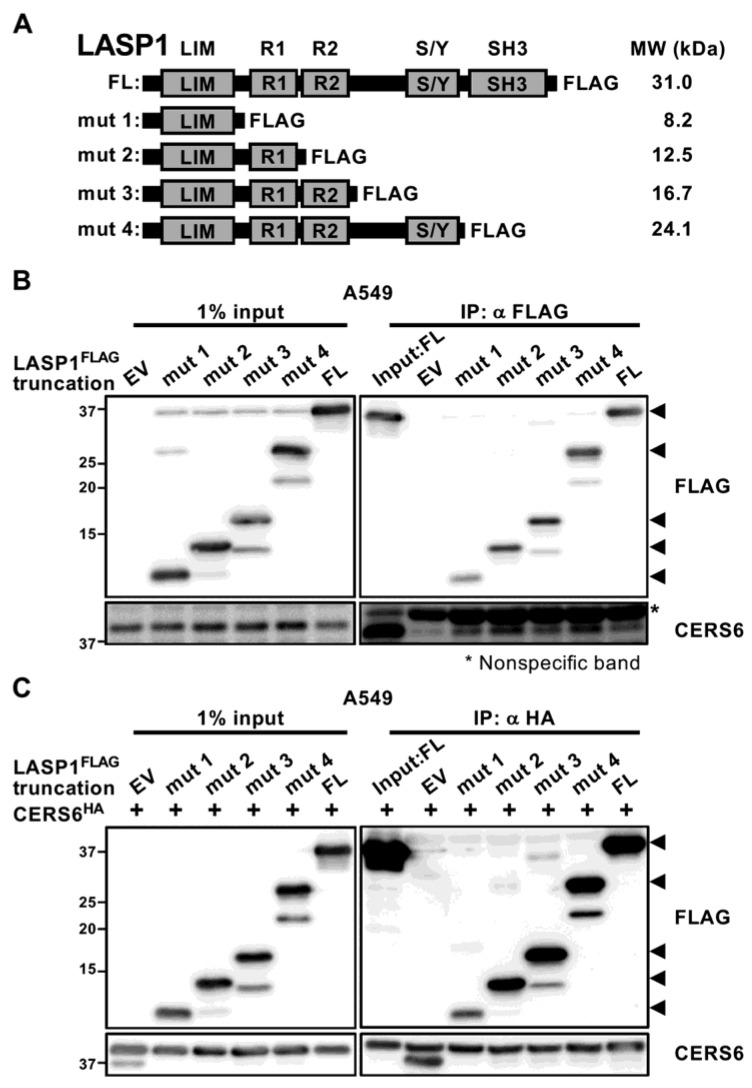
LIM domain of LASP1 required for CERS6 interaction. (**A**) Schematic illustration of C-terminal FLAG-tagged LASP1. The full-length (FL) and four truncated mutants (mut1-4) are shown. Predicted molecular weights are indicated. (**B**) IP-WB with an anti-FLAG antibody to precipitate LASP1-FLAG and demonstrate CERS6 interaction. EV, empty pCMV vector. Arrowheads: predicted positions for full-length and truncated LASP1. * Nonspecific band. (**C**) CERS6-HA was overexpressed in A549 cells and each truncated LASP1-FLAG mutant, then IP was performed using an anti-HA antibody. EV, empty pCMV vector. Arrowheads: predicted positions for full-length and truncated LASP1. Original western blots are presented in Appendix A.

**Figure 3 cancers-15-02781-f003:**
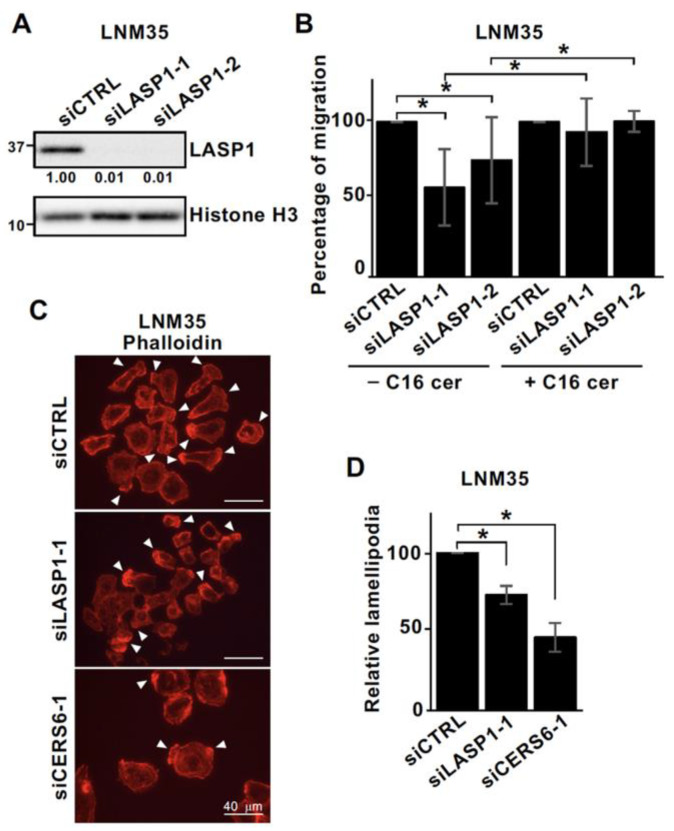
LASP1 and CERS6 in the same axis, with a possible contribution to the promotion of cell migration. (**A**) WB analyses of LNM35 cells treated with siCTRL, siLASP1-1, and siLASP1-2 were performed. Relative values to histone H3 are indicated. (**B**) LNM35 cells were treated with siCTRL, siLASP1-1, or siLASP1-2; then, cell migration activity was determined in the presence or absence of 1 μM of C16 ceramide (C16 cer). At least 100 cells were counted and the results are shown as the mean ± SD value relative to that of siCTRLs. * *p* < 0.05, six independent experiments. (**C**) LNM35 cells were treated with siCTRL, siLASP1-1, or siCERS6-1, then fixed and stained with phalloidin. Scale bar = 40 μm. Arrowhead: lamellipodia. (**D**) Quantitative results are shown as values relative to siCTRL. At least 200 cells were counted and the results are shown as the mean ± SD. * *p* < 0.05, three independent experiments. Original western blots are presented in Appendix A.

**Figure 4 cancers-15-02781-f004:**
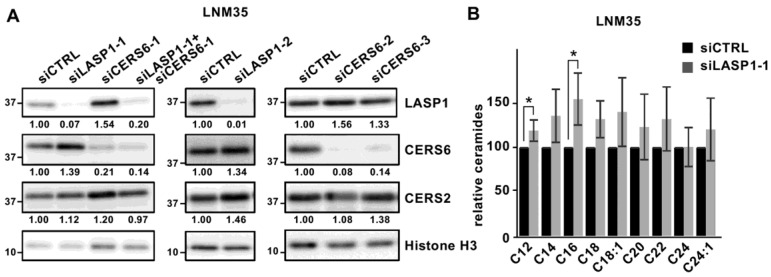
LASP1 and CERS6 modulated counterpart protein levels. (**A**) WB analysis of LNM35 cells was performed under knock-down conditions, as indicated. Relative values to histone H3 are shown. (**B**) Using LC-MS/MS, ceramide amounts were determined in siCTRL-treated (black) and siLASP1-1-treated (gray) LNM35 cells (n = 3). Quantitative values are shown as relative to siCTRL. * *p* < 0.05. Original western blots are presented in Appendix A.

**Figure 5 cancers-15-02781-f005:**
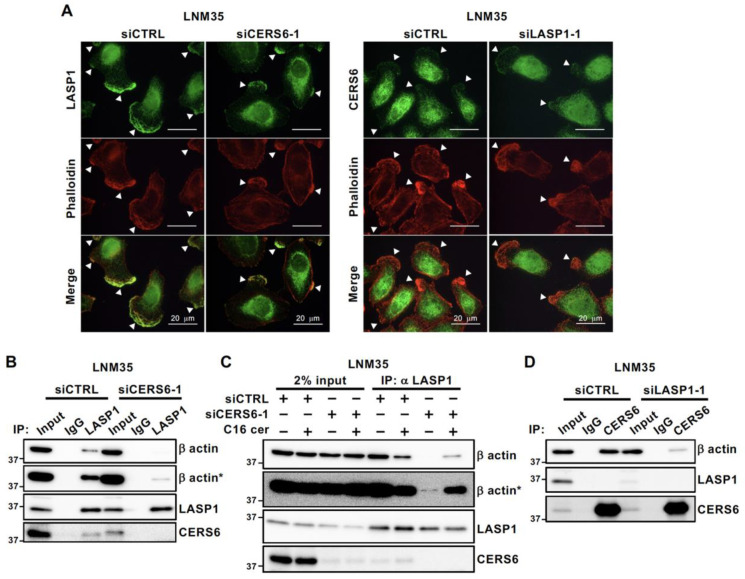
Association of β actin with LASP1 and CERS6. (**A**) Immunofluorescent staining of phalloidin and LASP1 or CERS6 at lamellipodia of LNM35 cells treated with siCTRL, siCERS6-1, or siLASP1-1. (**B**) IP-WB analysis was performed with the anti-LASP1 antibody to evaluate the interaction of LASP1–β actin. * Long exposure. (**C**) IP-WB analysis was performed with the anti-LASP1 antibody in the presence or absence of C16 ceramide (C16 cer) to evaluate interactions with LASP1–β actin. * Long exposure. (**D**) IP-WB analysis was performed with the anti-CERS6 antibody. Original western blots are presented in Appendix A.

**Figure 6 cancers-15-02781-f006:**
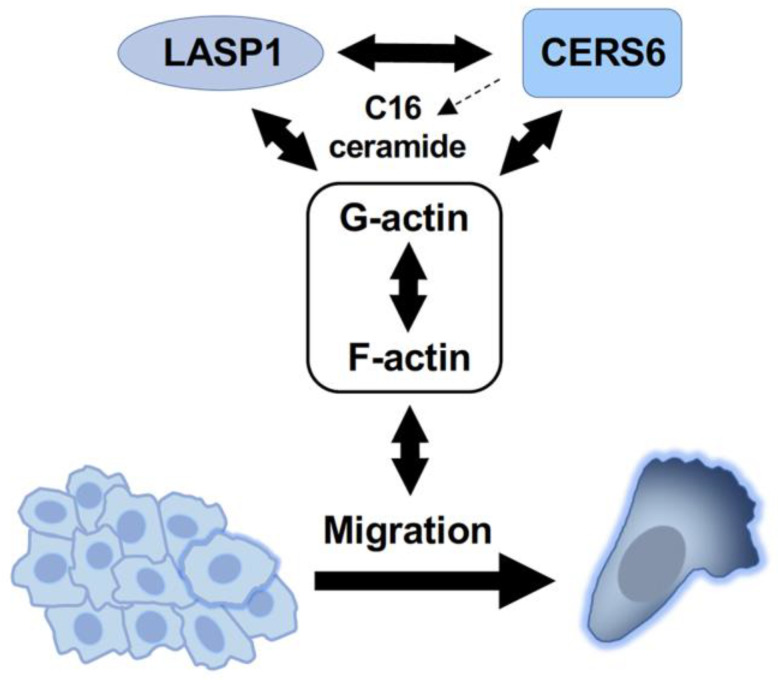
Schematic illustration showing a ternary complex, lamellipodia formation, and cell migration.

## Data Availability

The MS raw data and analysis files have been deposited in the ProteomeXchange Consortium (http://proteomecentral.proteomexchange.org, accessed on 14 May 2023) via the jPOST partner repository (http://jpostdb.org) with the data set identifier PXD042064.

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
