# Peer review of "LASP1, CERS6, and Actin Form a Ternary Complex That Promotes Cancer Cell Migration"

_cancers, 2023, doi:10.3390/cancers15102781_

Round 1
Reviewer 1 Report
Hello Authors,
I find your paper and finding interesting for the reader of this journal and will overall give benefit to large research community. However I feel if you could add few details as below, this would give reader more clarity. Also I feel you could show dataset with CRISPR KO of each gene and show change in gene expression and also see the change in expression using WB to overall compare to relation between LASP1 and CERS6. This would be even more beneficial and clear evidence.
LASP1 (LIM and SH3 domain protein 1) is a cytoskeleton-associated protein involved in cancer cell migration and invasion. CERS6 (ceramide synthase 6) is an enzyme that regulates ceramide synthesis, which is a component of the cell membrane that plays a role in cell signaling and apoptosis. Actin is a protein that forms the cytoskeleton, a network of fibers that provides structural support to cells.
It has been suggested that LASP1, CERS6, and actin can form a ternary complex, meaning a complex of three different proteins, which promotes cancer cell migration. This is thought to occur by regulating the dynamics of the cytoskeleton, allowing the cancer cells to move and invade surrounding tissue. However, more research is needed to fully understand the mechanism by which this ternary complex promotes cancer cell migration and its potential as a target for anti-cancer therapies.
-------------------------------------------------------------------------------------
Reviewer's opinion after reviewing the revised version:
Hello Editors, After completing the through read of the revised version submitted by the authors for the paper, "LASP1, CERS6, and actin form ternary complex that promotes cancer cell migration”, I believed the authors have made significant changes to the original version and provided sufficient evidence to support their hypothesis. With this revised version I believe this paper should be accepted in the present form. This would paper would be great insight for the readers of the journal and for the biotech research community. Best, Atiq Nurani
Author Response
"Please see the attachment."

Reviewer 2 Report
The authors investigated the possible direct interaction between the metastasis- and poor cancer prognosis-related CERS6 enzyme and LASP1protein involved in the cancer cell migration and invasion. This physical connection was demonstrated to occur through the LIM domain of LASP1. The close co-localization of the two proteins in lamellipodia and their co-immunoprecipitation with actin suggested that the CERS6-LASP1 interaction may affect the migration of cancer cells.
Major comments:
1. Figure 2A: Please explain the role and structure of R1, R2, S/Y domains in the text, and add the real or predicted molecular mass of truncated LASP1 proteins.
2. Figure 2B: In the opinion of the authors, what could be the band consistently appearing under mut2, mut3 and mut4 on the FLAG immunoblot?
3. Figure 2C: I disagree that Figure 2C is the opposite of the experiment in Figure 2B. In my opinion, this is merely a repeat of Figure 2B in cells transfected with HA-labeled CERS6 instead of endogenous CERS6. The experiment from the opposite direction could be if endogenous or overexpressed CERS6 is co-immunoprecipitated with truncated LASP1-FLAG protein. Have you conducted such experiments?
4. If the LIM domain of truncated LASP1 is improperly folded, would it make sense to make this domain non-functional by e.g. mutagenesis, to see if the 3D structure of the protein can be preserved and thereby obtain direct evidence of the role of the LIM domain in CERS6 binding.
Minor comments:
1. There are many places in the manuscript with text highlighted in red that should be removed.
2. A short paragraph about the role, structure and function of the LIM domain should be inserted into Introduction section.
3. Figure 1B and C: What was the reason for using the CLIP method for the results shown in Figure 1B and the IP-WB for those shown in Figure 1C?
4. For a better understanding, it would be worthwhile to represent the intensity values in Figure 4A and Figure S5A in the form of a bar graph.
Author Response
"Please see the attachment."

Reviewer 3 Report
In this manuscript Niimi et al, propose that a ceramide synthase associates with actin by forming a ternary complex with LASP1 to promote lung cancer metastasis. Although the proposed hypothesis is interesting some alterations are required for the manuscript to be considered for publication.
Major concerns:
Figure 1.
- please indicate on the graph from A. and the corresponding legend the unit on the basis of which the MS quantification was made (respectively PSM, that authors mentioned only in the material and method section), it would be clearer and explain the „bait” position from the bar graph showed in A. It does not correlate with figure 1B since CERS6 should be the most abundant protein identified.
-the same indication for supplementary MS table -Table S2 – the information is insufficient for non-specialists to understand the experimental design and also please specify why the difference of abundance ratio presented in the table is so big between the two replicates for some of the identified candidates.
- please load the original MS data in a repository platform and indicate the link, password in the manuscript.
-please introduce more details in the figure legend for point B and C about the cell line used. For Co-IP experiment the same cell line system was used, with the HA-CERS6 construct overexpressed? How was performed overexpression, as for mutant with electroporation?
Figure 2.
-the authors should explain better why they have concluded that mutant 5 lacking the LIM domain is not folded since no stability experiment has been conducted (eg. Cycloheximide chase). Moreover the expression level of the mutant is reasonable compared to mutant 5 in the same figure (input of S2B); please also indicate the number of replicates for the experiments performed.
Figure 3.
-which test was used to assess the statistical significance at the point B? Comparing more than two conditions ANOVA should be used assuming equal sample variance or an alternative non-parametric test. The error bars are quite large for some conditions to obtain significance. For example compare siCTR and siLASP1-2. How many experiments were performed? Please indicate this for each figure (n=…) and adjust the statistical comparison accordingly.
- how was lamellipodia defined for quantification purposes (eg: as protrusions/cell, as co-localization coefficient), same for figure S1
Figure 4.
-please add in A. quantification error bars and statistical analysis to assess reproducibility and significance.
-it is of more interest to quantify each ceramide relative to siCTRL and not all the others to C16. This would stress out if there are any additional ceramides impacted by siLASP1-1 treatment. Also a gapped y axis should be used as most of the ceramide species are barely visible. Does siLASP1-2 shows the same behavior? This should be also mentioned.
Minor:
1. Figure 1D: Please calculate specific coefficients to assess the degree of co-localization.
2. Figure 3B: Are there any variation for siCTRL in –C16 cer/ +C16 cer? This should be assessed statistically also and mentioned in the text.
3. Overall written English should be revised
Author Response
"Please see the attachment."

Round 2
Reviewer 1 Report
I believe this paper should be accepted in the present form. This would paper would be great insight for the readers of the journal and for the biotech research community.
Author Response
"Please see the attachment."

Reviewer 2 Report
The authors have answered my questions satisfactorily and made the necessary changes to the manuscript. In its present form, the manuscript is suggested for acceptance.
Author Response
"Please see the attachment."

Reviewer 3 Report
Figure 1: My initial suggestion was to mention in the figure legend and on the axis what numerical values from MS data were used to obtain the results in Figure 1A, not to remove the term PSM. On the contrary, if the authors analyzed the MS data based on PSMs then this should be clearly mentioned along the manuscript (the materials and methods section, figure legend and the results section), not to remove. This jus confuses the reader. Thus the authors should be more transparent when explaining how they obtained the data on which their manuscript conclusions are based.
Table S2: Same: please explain based on what you calculate abundance.
Please double-check the deposited data. The raw files from one replicate are empty.
Figure 3: As mentioned initially, when comparing multiple treatments ANOVA should be used (if all the data assumptions are met, or alternative a non-parametric alternative) not a test meant for comparison for a single treatment. Thus, I strongly suggest the authors to re-analyze their statics accordingly.
Author Response
"Please see the attachment."
